# Task Scheduling Based on a Hybrid Heuristic Algorithm for Smart Production Line with Fog Computing

**DOI:** 10.3390/s19051023

**Published:** 2019-02-28

**Authors:** Juan Wang, Di Li

**Affiliations:** School of Mechanical and Automotive Engineering, South China University of Technology, Guangzhou 510641, China; wjhao456@163.com

**Keywords:** fog computing, task scheduling, smart manufacturing, hybrid heuristic (HH) algorithm

## Abstract

Fog computing provides computation, storage and network services for smart manufacturing. However, in a smart factory, the task requests, terminal devices and fog nodes have very strong heterogeneity, such as the different task characteristics of terminal equipment: fault detection tasks have high real-time demands; production scheduling tasks require a large amount of calculation; inventory management tasks require a vast amount of storage space, and so on. In addition, the fog nodes have different processing abilities, such that strong fog nodes with considerable computing resources can help terminal equipment to complete the complex task processing, such as manufacturing inspection, fault detection, state analysis of devices, and so on. In this setting, a new problem has appeared, that is, determining how to perform task scheduling among the different fog nodes to minimize the delay and energy consumption as well as improve the smart manufacturing performance metrics, such as production efficiency, product quality and equipment utilization rate. Therefore, this paper studies the task scheduling strategy in the fog computing scenario. A task scheduling strategy based on a hybrid heuristic (HH) algorithm is proposed that mainly solves the problem of terminal devices with limited computing resources and high energy consumption and makes the scheme feasible for real-time and efficient processing tasks of terminal devices. Finally, the experimental results show that the proposed strategy achieves superior performance compared to other strategies.

## 1. Introduction

With the rapid development of emerging technologies such as the Internet of Things (IoT) [1], big data [2] and cloud computing [3], the industrial revolution has entered the so-called stage 4.0, and manufacturing modes have also entered the intelligent category [4]. Emerging technologies are widely used in the intelligent plant; in particular, a large amount of IoT equipment is deployed in the intelligent plant, analysing and processing enormous amounts of data that introduce challenges to cloud computing. Considering the disadvantages of cloud computing, fog computing is used to solve the processing of real-time tasks in the Industrial IoT [5]. The main difference between fog computing and cloud computing is that fog computing can provide low latency computing services for terminal devices, which is decided by the fog nodes deployed at the location. Fog nodes are usually deployed around the terminal devices, and often through one jump, they can complete data forwarding, greatly reducing the data transmission delay; however, this advantage cannot be achieved with cloud computing.

Rely on its own advantages, fog computing plays a great role in the field of smart manufacturing. The performance of fault detection and state analysis of devices in production line can be improved through a wealth of computational and storage services provided by fog computing [6]. Efficient manufacture inspection systems with fog computing can be implemented [7]. A fog-based solution is proposed in [8] for real-time monitoring and data processing in manufacturing. By using fog computing technology, an enhancing smart maintenance management solution is presented in [9]. However, among the existing works few focus on task scheduling with fog computing in the smart manufacturing area. Despite the fact fog computing provides many convenient computing services for manufacturing task processing [10], however, fog computing still faces many challenges in manufacturing applications [11,12], particularly since the fog service ability is ultimately limited, and each fog node has a strong heterogeneity, such as differences in computing ability, storage capacity and communication ability. The task requests from terminal devices also have very strong heterogeneity, such as real-time requirements, energy consumption requirements, etc. Therefore, determining the approach of task scheduling among different fog nodes to minimize the delay and energy consumption is the purpose of this paper. As well as we know, task scheduling is an NP-hard problem, it is also a very challenging problem. Therefore, a hybrid heuristic algorithm to solve the task scheduling problem has been a research hotspot of scholars worldwide. The hybrid heuristic algorithm which combines the advantages of a variety of heuristic algorithm, the accuracy of results and the optimization process are both improved. The main contributions of this paper can be summarized as follows:
(1)We design a distributed fog computing system architecture for smart production line. The fog computing system model and task scheduling-related mathematical model are established. The task scheduling objective function is given, whose goal is to minimize the delay and energy consumption.(2)A hybrid heuristic algorithm is proposed, which combines the improved particle swarm optimization (IPSO) algorithm and the improved ant colony optimization (IACO) algorithm to provide a hybrid heuristic algorithm for the task scheduling problem.(3)We establish a fog computing based smart production line simulation environment, and the task scheduling strategy proposed is compared with other strategies, then using three performance metrics for experimental verification, achieving results that prove the effectiveness of the strategy.

The rest of this paper is organized as follows: the related work is presented in Section 2. The system model, task scheduling-related mathematical model and the task scheduling objective function are established in Section 3. A task scheduling strategy is put forward in Section 4, and the task scheduling problem is solved by a hybrid heuristic algorithm, which combines the advantages of the IPSO with the IACO. Experimental verification is given in Section 5, the task scheduling strategies are compared using three performance metrics, the simulation results showing that the proposed scheduling strategy is superior to other strategies. Finally, Section 6 concludes the paper.

## 2. Related Work

In recent years, many scholars worldwide have conducted research on fog computing, and the main research directions are focused on the definition [13,14], architecture [15,16,17], application [18,19,20], computing offloading [21,22,23] and task scheduling [24,25,26]. Based on the terminal equipment and its requirement of real-time performance and energy consumption, the task scheduling of the fog computing mode is a necessary research hotspot. The task scheduling research on fog computing is still at the preliminary stage. Yin et al. [6] introduced fog computing in the intelligent manufacturing environment in which the container virtual technology was adopted, and through the task scheduling and resource allocation to ensure the real-time task performance, the reallocation mechanism to further reduce the computing delay of the task was accomplished. Yang et al. [25] studied the task scheduling in a homogeneous fog network; they put forward a novel delay–energy balance task scheduling algorithm, and reduced the average service time delay and delay jitter of minimizing the overall energy consumption at the same time. Pham et al. [26] formulated the task scheduling problems in a cloud-fog environment, and proposed a heuristic-based algorithm. Chekired et al. [27] designed a multilayer fog computing architecture, in which they calculated the priority of IoT data and task requests; then, according to the priority conduct rank, high priority tasks that required fast deployment were accommodated by using two priority queuing models to complete industrial network scheduling and analysis of the data. Liu et al. [28] investigated a joint optimization algorithm of scheduling multiple jobs and a lightpath provisioning for minimizing the average completion time in a fog computing micro datacenter network. Bittencourt et al. [29] studied mobility-aware application scheduling in fog computing. Zeng et al. [30] designed an efficient task scheduling and resource management strategy with a minimized task completion time for promoting the user experience. Deng et al. [31] studied workload scheduling towards worst-case delay and optimal utility for a single-hop Fog-IoT architecture. A workload dynamic scheduling algorithm is proposed, which can maximize the average throughput utility while guaranteeing the worst-case delay of task processing. Chen et al. [32] applied fog computing technologies for enhancing the vehicular network, and two dynamic scheduling algorithms are proposed based on the fog computing scheme for the data scheduling in vehicular networks, in which these algorithms can dynamically adapt to a changeable network environment and achieve a benefit in efficiency. Zhao et al. [33] proposed a fog-enabled multitier network architecture which can model a typical content delivery wireless network. A new fog enabled multitier operations scheduling approach based on Lyapunov optimization techniques is developed to decompose the original complicated problem into two operations across different tiers. Extensive simulation results show the algorithm is fair and efficient. Wang et al. [34] designed a fog computing-assisted smart manufacturing system, and a Software-Defined IIoT system architecture based on fog computing was set up in a smart factory. An adaptive computing mode selection method is proposed, and simulator results show that this method can achieve real-time performance and high reliability in IIoT. Ni et al. [35] proposed a resource allocation strategy for fog computing based on priced timed Petri nets (PTPN). The strategy comprehensively considers the price cost and time cost to complete a task, and constructs the PTPN models of tasks in fog computing in accordance with the features of fog resources. The algorithm that predicts the task completion time is presented. The method of computing the credibility evaluation of the fog resource is also proposed. A dynamic allocation algorithm of fog resources is given. Simulation results demonstrate the proposed algorithms can achieve a higher efficiency than static allocation strategies in terms of task completion time and price.

The current main heuristic algorithms are the genetic algorithm (GA), particle swarm optimization (PSO) algorithm, ant colony optimization (ACO) algorithm, simulated annealing (SA) algorithm, and so on. The heuristic algorithms mainly mimic natural phenomena or rules and have a very good self-organizing, self-learning and adaptive ability, obtaining global optimal solutions with good robustness, therefore, these are widely used in complex problem solving. To obtain a better scheduling performance, many scholars have improved heuristic algorithms. In [36], a novel architecture is proposed, in addition to a task scheduling algorithm based on a dynamic scheduling queue algorithm and particle swarm optimization algorithm, and this algorithm fully considers the dynamic characteristic of the cloud computing environment, with experimental results that show the architecture can effectively achieve good performance, load balancing and improvements in resource utilization. In [37], an improved genetic algorithm for the task scheduling strategy was proposed. Moreover, in some literature, combinations of multiple single algorithms to form a hybrid algorithm for solving the task scheduling problem are proposed. Rahbari et al. [38] proposed a safety-aware scheduling scheme based on a hybrid heuristic algorithm in the fog computing environment. Dai et al. [39] proposed a task scheduling algorithm based on genetic and ant colony optimization algorithms in cloud computing. Ref. [40] focuses on the task scheduling and resource management problems in the cloud computing environment, and a hybrid bio-inspired algorithm was proposed to implement task scheduling. Ref. [41] put forward a mixed average minimum and max-min algorithm in cloud computing for task scheduling. To solve the problem of cloud computing load balancing, the literature [42] proposed an improved load balanced min-min algorithm using a genetic algorithm. In addition, the traditional scheduling algorithms, such as the first come first server (FCFS) algorithm, the round-robin (RR) scheduling algorithm, the max-min, min-min and other algorithms, which are relatively simple and are often used for performance comparison with other algorithms. There are many works in the literature about task scheduling algorithms that have been published; however, most of them are applied in cloud computing, while the literature of task scheduling algorithms in fog computing is sparse. Therefore, this paper focuses on the task scheduling strategy in fog computing.

## 3. System Model and Problem Formulation

In this section, a smart manufacturing system architecture based on fog computing is set up, we describe the system model. The delay mathematic model and energy consumption mathematic model of task processing under different fog nodes are formulated.

### 3.1. System Architecture and System Model

More and more delay sensitive and computing intensive tasks need to be processed in smart manufacturing scenarios. Fog computing can provide real-time computing services for terminal devices by closely deploying fog nodes. However compared to a cloud server, the computing ability of fog nodes are limited and different, so how to use the computing resources of fog nodes effectively is the purpose of this study. First of all, we establish a smart production line system architecture based on fog computing. As shown in Figure 1, the system architecture consists of four layers: terminal device layer, fog computing layer, cloud computing layer and application layer.

*Terminal device layer*: The terminal device layer mainly includes production line processing devices (manipulators, motors, CNCs), transmission equipment (conveyor belts, AGV), sensing devices and a variety of handheld terminal devices. Processing devices are mainly used to complete the product formulation, sensing devices are responsible for processing during the process of data collection, and handheld terminals help visualize the results.

*Fog computing layer*: The fog computing layer is located in middle position between the terminal device layer and cloud computing layer. Fog nodes which are deployed on the edge of the network mainly provide services for delay sensitive tasks. In smart production lines, the fog nodes refer to switches and routers as well as some special servers. For some common computing equipment, such as intelligent sensors, intelligent processing equipment, intelligent multimedia devices can also be used as fog nodes. Fog nodes have a certain computation capacity, communication ability and storage capacity, and these work between the cloud and the terminal device. Fog computing has remedied the high latency of cloud computing, thus the quality of service of real-time application can be guaranteed. However, the fog nodes’ ability is limited after all, the fog nodes also have a strong heterogeneity and dynamism in the face of vast amounts of real-time data processing, so how to rapid and efficient data analysis is the main goal of the fog computing.

*Cloud computing layer*: The cloud computing layer contains the cloud data center, cloud storage, cloud computing equipment, providing a remote service to intelligent production lines. High-performance computing equipment and large capacity storage devices which can transmit, calculate and store all kinds of huge data from terminal equipment through the control center of coordination management to provide comprehensive, high quality service for terminal users. Due to its remote deployment, cloud computing can’t complete real-time data processing and analysis, so cloud computing can’t meet the QoS demands of real-time tasks.

*Application layer*: The applications of the smart production line are product detection, fault diagnosis, device maintenance, real-time monitoring, inventory management, etc.

The heterogeneous task processing flow in the fog environment of a production line is shown in Figure 2. In a smart factory, *G* is a terminal device set, *G* = {*g*_1_, *g*_2_, *g*_3_, …, *g_i_*, …, *g_n_*}, where *i* denotes the serial number of the terminal device. The attributes of the terminal device *g_i_* are described by *P_g_* = {*p_ir_*, *p_ie_*, *a_i_*, *w_it_*, *w_ie_*, *E_il_*}, where *p_ir_* denotes the transmission power of *g_i_*, *p_ie_* denotes the idle power of *g_i_*, and *a_i_* is a binary variable, which is a mobile symbol for *g_i_*. *w_it_* denotes the delay weight of *g_i_*, and *w_ie_* denotes the energy consumption weight of *g_i_*. If *a_i_* = 1 denotes *g_i_* is a mobile device, the energy of the mobile terminal equipment is limited. In the process of task scheduling, we need to consider the energy consumption problem, so we set *w_it_* = 0.7, *w_ie_* = 0.3; if *a_i_* = 0 denotes *g_i_* is a static device, the energy of the static terminal equipment can be considered infinite, so we set *w_it_* = 1, *w_ie_* = 0. *E_il_* denotes the residual energy of *g_i_*.

For the task set *Φ_F_*, *Φ_F_* = {*I*_1_, *I*_2_, *I*_3_, …, *I_i_*, …, *I_n_*}, *I_i_* denotes the task request of terminal equipment *g_i_*, the attribute of *I_i_* is described by *P_I_* = {*D_i_*, *θ_i_*, *T_i_*_,*max*_, *T_i_*_,*exp*_}, *D_i_* denotes the data size of task *I_i_*, *θ_i_* denotes the computation intensity of task *I_i_*, *T_i_*_,*max*_ denotes the maximum tolerate time of task *I_i_*, *T_i_*_,*exp*_ denotes the expected completion time of task *I_i_*.

For the fog nodes set, *F* = {*F*_1_, *F*_2_, *F*_3_, …, *F_j_*, …, *F_m_*}, *j* denotes the number of fog nodes, the attribute of *F_j_* is described by *P_F_* = {*C_j_*, *K_j_*, *B_j_*}, *C_j_* denotes the computing resource of fog node *F_j_*, *K_j_* denotes the storage capacity of fog node *F_j_*, and *B_j_* denotes the network bandwidth of fog node *F_j_*.

We assume that time is slotted, and denote the time slot length and the time slot index set by Δ*τ* and *φ_t_* = {0, 1, 2, …}, respectively.

### 3.2. Delay Model and Energy Consumption Model

The task scheduling problem in fog computing can be described as *n* independent tasks assigned to *m* fog nodes, in which the goal of the task scheduling is to minimize the time cost and energy consumption cost through a reasonable allocation.

At slot *t*, *t* ∈ *φ_t_*, *s_ij_*(*t*) denotes the allocation relationship between task *I_i_* and fog node *F_j_*, *s_ij_*(*t*) ∈ {0,1}, if task *I_i_* is executed at fog node *F_j_*, *s_ij_*(*t*) = 1; otherwise, *s_ij_*(*t*) = 0.

The completion time includes the task execution time *te_ij_* and the transmission time, at slot *t*, the execution time of task *I_i_* at fog node *F_j_* can be expressed as:
*TE_ij_*(*t*) = *D_i_*(*t*)*θ_i_*/*C_j_*(*t*),(1)

*D_i_*(*t*) denotes the data size of task *I_i_*, *θ_i_* is the computing intensity of task, *C_j_*(*t*) denotes the computing resource of fog node *F_j_*.

At slot *t*, the transmission time is sending task *I_i_* to fog node *F_j_*:
*TR_ij_* = *D_i_*(*t*)/*r_ij_*(*t*),(2)
where *r_ij_*(*t*) is the transmission rate between terminal device *g_i_* and fog node *F_j_*:(3)$$r_{ij}(t)=\omega (t)*\log_{2}(1+\frac{h(t)*p(t)}{\sigma }),$$
where *ω*(*t*) denotes the network bandwidth, *σ* is the noise power, *h*(*t*) denotes the channel power gain, *p*(*t*) is the transmission power.

Therefore the total delay can be expressed as:(4)$$T_{ij}(t)=TR_{ij}(t)+TE_{ij}(t),$$

From the perspective of terminal devices, the energy consumption on terminal device *g_i_* is divided into two parts, one part is transmission energy consumption; the other part is waiting energy consumption:(5)$$E_{ij}(t)=TR_{ij}(t)*p_{ir}(t)+TE_{ij}(t)*p_{ie}(t),$$

### 3.3. Problem Formulation

Task scheduling in fog computing mode is designed to allocate multiple tasks to multiple fog nodes according to a certain scheduling strategy, and, as far as possible, to meet the task requests of terminal equipment while simultaneously reducing the task completion time and the energy consumption of the terminal equipment.

To simplify the problem complexity and reduce the difficulty of solving the problem, the following assumptions are put forward: each task is independent and there is no constraint relationship among tasks; each task can only be allocated to a fog node and none of the tasks are allowed to allocate repeatedly. The task in the computation process does not consider the impact of the mobility of the terminal equipment. All the fog nodes are static and task in the process of execution cannot be interrupted.

The objective function mainly considers the overhead of the terminal equipment to perform all tasks, including the completion time of all tasks and the energy consumption of all the mobile terminal equipment; therefore, the overhead of fog nodes will not be considered here. The objective function of task scheduling in the fog computing with delay and energy consumption constraints is formulated as follows:(6)$$f=\min\sum_{i=1}^{n}\{w_{it}\sum_{j=1}^{m}\left[s_{ij}(t)*T_{ij}(t)]+w_{ie}\sum_{j=1}^{m}[s_{ij}(t)*E_{ij}(t)]\right\}$$
s.t. (C1): *s_ij_*(*t*) ∈ {0,1}, *i* = 1, 2, …, *n*; *j* = 1, 2, …, *m*,(C2): *s_i_*_1_(*t*) + *s_i_*_2_(*t*) + … + *s_im_*(t) = 1, *i* = 1, 2, …, *n*,(C3): *s_ij_*(*t*)**T_ij_*(*t*) ≤ *T_i_*_,*max*_, *i* = 1, 2, …, *n*; *j* = 1, 2, …, *m*,(C4): *s_ij_*(*t*)**E_ij_* (*t*)≤ *E_il_*, *i* = 1, 2, …, *n*; *j* = 1, 2, …, *m*,(C5): $\left\{ \begin{array}{l} w_{it}=0.7\\ w_{ie}=0.3 \end{array}\right.$, *a_i_* = 1, *i* = 1, 2, …, *n*,(C6): $\left\{ \begin{array}{l} w_{it}=1\\ w_{ie}=0 \end{array}\right.$, *a_i_* = 0, *i* = 1, 2, …, *n*,
where (C1) and (C2) are the constraints on the task scheduling decision, namely, that each task can only be allocated to a fog node; (C3) are the delay constraints on each task; (C4) are the energy consumption constraints on each task; (C5) are the delay weight constraints on each terminal equipment; and (C6) are the energy consumption weight constraints on each terminal equipment.

## 4. Task Scheduling Algorithm Design

The task scheduling is a multi-objective nonlinear combinatorial optimization problem. There are many multi-variables and multi-constraints in the objective function, therefore, it is difficult to find the optimal solution through the polynomial method. A hybrid heuristic algorithm is designed which combines the improved discrete particle swarm optimization algorithm with the improved ant colony optimization algorithm, and a task scheduling strategy about fog computing based on the hybrid heuristic algorithm is proposed.

### 4.1. Improved Particle Swarm Optimization Algorithm

The particle swarm optimization (PSO) algorithm provides a quick and efficient choice for solving complex optimization problems, but because the parameters of the task scheduling problem in fog computing are discrete, the standard particle swarm optimization algorithm cannot be used to solve the optimization. Therefore, the standard particle swarm optimization algorithm requires discretization processing, and to redefine the particle’s position and speed, the discrete particle swarm optimization algorithm is adopted to solve the task scheduling problem.

#### 4.1.1. Discrete Particle Swarm Initialization

A direct binary encoding method is adopted, therefore the particle’s location information is represented by 0 and 1, and the velocity of the particles is in the interval [0, 1]. In a discrete particle swarm, the position of each particle represents a possible task scheduling scheme, and according to the task allocation decisions *s_ij_*(*t*), the position of particle *i* can be simplified as *s_ij_*, *s_ij_* ∈ {0,1}, *i* = 1, 2, …, *n*; *j* = 1, 2, …, *m*. Similarly, the speed of the particle *i* can be simplified as *v_ij_*, *v_ij_* ∈ [−*v_max_*_,_
*v_max_*], *i* = 1, 2, …, *n*; *j* = 1, 2, …, *m*.

#### 4.1.2. Updating Position and Speed of the Particle

To improve the performance of the basic particle swarm optimization algorithm, we introduce the adaptive inertia weight *ω* and the contraction factor *η*. The inertia weight *ω* plays an important role for the particle’s search ability. The contraction factor *η* not only can accelerate the convergence speed of the particles but can also improve the accuracy of the algorithm. The inertia weight *ω* and the contraction factor *η* can improve the performance of the particle swarm optimization algorithm, so the combination of both is represented in the particle velocity updating equation:(7)$$v_{ij}^{k+1}=\eta (\omega v_{ij}^{k}+c_{1}r_{1}(p_{ij}^{k}-s_{ij}^{k})+c_{2}r_{2}(g_{ij}^{k}-s_{ij}^{k}))$$

To represent the position vector of the particles *s_ij_* values as binary variables, the sigmoid function is introduced:(8)$$Sig(v_{ij}^{k+1})=\frac{1}{1+\exp(-v_{ij}^{k+1})}$$

The particle position update formula is:(9)$$s_{ij}^{k+1}=\{\begin{array}{c} 1,r_{3}<Sig(v_{ij}^{k+1})\\ 0,r_{3}\geq Sig(v_{ij}^{k+1}) \end{array}$$
where *k* is the serial number of the iteration, *p_ij_* is the best location for the particle individual of the *j* vector, *g_ij_* is the best location for particle groups of the *j* vector, $\nu_{ij}^{k}$ is the current velocity of the *k* generation particle, *c*_1_ and *c*_2_ are accelerating factors, and *r*_1_, *r*_2_, *r*_3_ are random numbers in [0, 1].

#### 4.1.3. Fitness Function

The fitness function is the objective function of the particle swarm algorithm, and through the fitness function, the particle swarm judges the stand or fall of the current position and speed, so that it constantly searches and updates the individual and global optimum and the fitness of some value for the target iteration until the termination of the algorithm. When the fitness function value is greater, the solution is better; according to Equation (6), the fitness function is shown in Equation (10):(10)$$fitness=\frac{1}{\min\sum_{i=1}^{n}\{w_{it}\sum_{j=1}^{m}\left[s_{ij}(t)*T_{ij}(t)]+w_{ie}\sum_{j=1}^{m}[s_{ij}(t)*E_{ij}(t)]\right\}}$$
where the objective function of task scheduling is the denominator of Equation (10), when the fitness function value is greater, the solution is better.

#### 4.1.4. Inertia Weight *ω* and Contraction Factor *η*

The inertia weight *ω* determines the search ability of the particles in the global and local search, which follows the premise that the greater the *ω* is, the stronger the global searching ability of the algorithm; otherwise, the local search ability of the algorithm is stronger. To improve the intelligence of the particle swarm optimization algorithm, many methods have been proposed regarding inertia weighting, such as: linear decreasing weight, weight decrease of linear differential and so on. This paper combines the inertia weight with the number of iterations, and the solution formula of the inertia weight is shown in Equation (11):(11)$$\omega =\{\begin{array}{cc} \omega_{\max}-\frac{(\omega_{\max}-\omega_{\min})\times k}{K_{\max}}, & k<0.7\times K_{\max}\\ \omega_{\min}+(\omega_{\max}-\omega_{\min})\times rand, & k\geq 0.7\times K_{\max} \end{array}$$
where *ω*_max_ is the maximum inertia weight, *ω*_min_ is the minimum inertia weight, *k* is the number of iterations, and *k*_max_ is the largest number of iterations. Through adjusting the inertia weight *ω*, the improved algorithm not only retains the original global search ability but also improves the local search ability. It can be seen from (11), that the inertia weight *ω* is changed dynamically with the increase of the iteration times, which guarantees that the algorithm has the opportunity to gain a larger inertia weight value later in the search and is prevented from falling into the local optimum.

The contraction factor and inertia weight factor have the same function, such that when the contraction factor is larger, the global search ability of the algorithm is enhanced, whereas the local search ability is enhanced. This contraction factor computation formula is as follows:(12)$$\eta =\frac{2}{\left|2-\varphi -\sqrt{\varphi^{2}-4\varphi }\right|}$$
where *ϕ* = c_1_ + c_2_, *ϕ* >4.

### 4.2. Improved Ant Colony Optimization Algorithm

The ant colony optimization (ACO) algorithm is a bionic optimization algorithm, inspired by biological evolution. The ant colony optimization algorithm is based on swarm intelligence, and the group size is usually large, the motion state of each individual in the group is random, and the process of the ant colony is to find the shortest path according to the nature of the proposed dynamic random search algorithm. This algorithm has the advantages of flexibility and robustness, and the dynamic environment system is used to solve combinatorial optimization problems. Therefore, to improve the efficiency of task scheduling and the satisfaction in fog computing, the ant colony optimization algorithm is used to solve the combinatorial optimization problem between the fog nodes and tasks.

Based on the ant colony optimization algorithm in the fog computing scenario, the task scheduling problem can be expressed by a directed acyclic graph (DAG). *G* = (*S*, *L*), *S* denotes the set for the task allocation nodes, *S* = {*s_ij_*(*t*) | *i* = 1, 2, ..., *n*, *j* = 1, 2, ..., *m*}. *L* denotes the path set, *L* = {*l_ij_* | *i* = 1, 2, ..., *n*, *j* = 1, 2, ..., *m*}. The ant selecting path *l_ij_* with the denoted task *I_i_* is assigned to the fog node *F_i_*, and the length of the path *l_ij_* is a computational overhead of task *I_i_* in fog node *F_i_*, where the computational overhead includes the delay cost and energy consumption costs, which follow the premise that the longer the length of the path *l_ij_* is, the larger the computational overhead, |*l_ij_*| = *f_ij_* = *T_ij_*(*t*) + *E_ij_*(*t*).

The total number of ants is *σ*, and each ant has the following features: the ant walking route is a directed acyclic graph, as shown in Figure 3, and all the ants start from the starting point for *S_S_*, then choose the next node according to the path transition probability. The path transition probability is a function of the path distance and the path pheromone amount. The legal walking route of each ant is shown in Figure 3, and to prevent the ant to have ever walked to the node, a tabu table is introduced. The tabu table of ant *k* is set to *tabu_k_*; after the ant *k* has walked to the node *s_ij_*, the node *s_ij_* will be added to the tabu table, and in the process where the ants are walking below it, *s_ij_* will not be chosen. When the ant *k* walks to the end point *S_E_*, the ant has completed its travel (all task scheduling), and the ants leave pheromone on the path to walk through.

#### 4.2.1. Improved Heuristic Information

The heuristic information of the ant colony algorithm can make the ants preferred path satisfy the objective function, namely, the expectations of task *I_i_* assigned to the fog node *F_j_*. If the computational overhead is larger, the *η_ij_* will be smaller, so the probability of the ant selecting the node is small. In the previous task scheduling application with the basic ant colony algorithm optimization design, the heuristic information *η_ij_* is equal to 1/*f_ij_*, and *f_ij_* denotes the computational overhead of task *I_i_* on fog node *F_j_*. However, this situation only considers the expenses of the current fog node without considering the influence of the global state of the heuristic information, thus reducing the efficiency and accuracy of the obtained global optimal solution. Here, therefore, the heuristic information should be improved, as shown in (13):(13)$$\eta_{ij}=\omega_{1}\frac{1}{f_{ij}}+\omega_{2}\frac{1}{\sum_{v=1}^{i-1}f_{vj}}$$
where *f_ij_* is the computational overhead of task *I_i_* on fog node *F_j_*, $\sum_{\nu =1}^{i-1}f_{\nu j}$ is the computational overhead of tasks that have chosen fog nodes. *ω*_1_ and *ω*_2_ are weight of the local computational overhead and the global computational overhead, respectively, *ω*_1_ + *ω*_2_ = 1, *ω*_1_, *ω*_2_ ∈ [0, 1].

#### 4.2.2. Improved Path Transition Probability

Some ants are placed on the start node *S_S_* to travel, and the probability of the *k*-th ant from the current routing to selecting a neighbouring path is determined by the pheromone *τ_ij_* and the heuristic information *η_ij_*. To avoid allowing the ant colony optimization algorithm to be prematurely trapped in a local optimum, the path transition probability is further improved. Therefore, a regulating factor *μ_ij_* is introduced to the path transition probability formula, and as the number of iterations increases, the factor is conducive to the ant selecting the pheromone of the smaller node, and avoiding some nodes’ pheromone where the phenomenon of rapid accumulation appears, thus guaranteeing that the ants can still search for a better solution in the later iterations and avoid having the algorithm encounter the premature phenomenon. The calculation formula of the path transition probability is shown below:(14)$$p_{ij}{}^{k}=\left\{ \begin{array}{l} \frac{{\left[\tau_{ij}\right]}^{\alpha }{\left[\eta_{ij}\right]}^{\beta }\mu_{ij}}{\sum_{i\in allowed_{k}}{\left[\tau_{ij}\right]}^{\alpha }{\left[\eta_{ij}\right]}^{\beta }\mu_{ij}},i\in allowed_{k}\\ 0,else \end{array}\right.$$
where *τ_ij_* is the residual pheromone of node *s_ij_*, *η_ij_* is the heuristic information of node *s_ij_*, *allowed_k_* denotes the node set allowed access by ant *k*, *allowed_k_* = *S* − *tabu_k_*. *α* is the pheromone weight of the path, *β* is the heuristic information weight of the path. *μ_ij_* is the regulatory factor of the path transition probability, and the computational formula is given as:(15)$$\mu_{ij}=e^{\left(-|\tau_{ij}-\tau |\right)}$$
where *τ* is the pheromone average value of all nodes for each iteration. From (15), the path transition probability adjustment factor *μ_ij_* is decided by *τ_ij_* and *τ*.

#### 4.2.3. Update of the Pheromone

After the ant *k* has selected node *s_ij_*, for which task *I_i_* is assigned to the fog node *F_j_*, the pheromone of node *s_ij_* needs to be updated since the pheromone update improves the convergence speed and precision of the ant colony algorithm. The local pheromone updating formula is shown in (16) and (17):(16)$$\tau_{ij}(t+1)=(1-\rho )\tau_{ij}(t)+\sum_{k=1}^{\sigma }\Delta \tau_{ij}{}^{k}$$
(17)$$\Delta \tau_{ij}{}^{k}=\left\{ \begin{array}{ll} \frac{Q}{f_{ij}{}^{k}}, & \mathrm{the}\;k\mathrm{th}\;\mathrm{ant}\;\mathrm{choose}\;s_{ij}\;\mathrm{at}\;t\\ 0, & else \end{array}\right.$$
where *ρ* is the pheromone volatility coefficient, 1 − *ρ* is the pheromone residual coefficient, *ρ* ∈ [0,1). Δ*τ_ij_^k^*(*t*) denotes the pheromone of ant *k* on node *s_ij_* at time *t*. *Q* is a constant, *f_ij_^k^* is the length of path *l_ij_* chosen by ant *k*:(18)$$\tau_{ij}(t+n)=(1-\rho )\tau_{ij}(t)+\sum_{k=1}^{\sigma }\Delta \tau_{ij}{}^{k}$$
(19)$$\Delta \tau_{ij}{}^{k}=\left\{ \begin{array}{ll} \frac{Q}{f^{k}}, & \mathrm{the}\;k\mathrm{th}\;\mathrm{ant}\;\mathrm{choose}\;s_{ij}\;\mathrm{at}\;\mathrm{current}\;\mathrm{iteration}\\ 0, & else \end{array}\right.$$
where Δ*τ_ij_^k^* denotes the pheromone of ant *k* on node *s_ij_* in the process of the current iteration, and *f^k^* denotes the path length that ant *k* has walked in the process of the current iteration.

### 4.3. Hybrid Heuristic Algorithm

The PSO algorithm and ACO algorithm are introduced in detail, and the two kinds of algorithms are improved. However, each algorithm has its own limitations. For example, the PSO algorithm has a fast search speed, but the accuracy is not high, and the ACO algorithm has high precision, but the search speed is low and so on. To solve the legacy problems of PSO and ACO, a HH algorithm can compensate for the shortage of the single heuristic algorithm, and the HH algorithm combines the advantages of the IPSO algorithm with the IACO algorithm. First, the fast convergence characteristics of the IPSO algorithm are used to collect the optimal solution, and then the optimal solution is the initial pheromone distribution of the IACO algorithm, so that the IACO algorithm is used for the optimal solution of task scheduling. The flowchart of the HH algorithm is shown in Figure 4.

The basic steps of the task scheduling strategy in fog computing based on the hybrid heuristic algorithm are as follows:Step 1:Define the task scheduling objective function in fog computing.Step 2:Set related parameters of the hybrid heuristic algorithm and the algorithm termination conditions.Step 3:Initialize the position and speed of the improved discrete particle swarm.Step 4:Calculate each particle’s fitness, and determine the particle’s individual optimal position and the global optimal position.Step 5:According to Equations (7) and (9), the velocity and position of the discrete particle are updated.Step 6:Judging whether the algorithm’s termination conditions are satisfied, output the task initial scheduling results of the fog computing mode, and turn to Step 7; otherwise, go to Step 4.Step 7:Referencing to the scheduling results of the improved particle swarm optimization algorithm to initialize the pheromone of the improved ant colony optimization algorithm.Step 8:Some ants are placed on the start node for travelling.Step 9:Each ant according to Equation (14) chooses the next node; according to Equations (16) and (17), update the local pheromone, and add the selected node to the task scheduling list.Step 10:After scheduling all the tasks, calculate the fitness value of the scheduling results according to the task scheduling list, and then according to Equations (18) and (19), update the global pheromone. Otherwise, go to Step 9.Step 11:Judging whether the algorithm’s termination conditions are met, output the task optimal scheduling results of the fog computing mode. Otherwise, go to Step 8.

## 5. Implementation and Simulation Results

In this section, simulations are conducted to evaluate the performance of the proposed task scheduling strategy. Both the simulation setup and performance metrics are described detail. Our strategy is compared with three other strategies from four different performance metrics, and the evaluation results verify that our strategy is superior to other strategies.

### 5.1. Simulation Setup

We develop the simulation platform and verify the proposed task scheduling strategy in the MATLAB environment. The fog computing system model and the task scheduling related model are set up, and the improved particle swarm optimization algorithm and the improved ant colony optimization algorithm are introduced to solve the task scheduling objective function. The system model parameters and simulation parameters are shown in Table 1.

### 5.2. Performance Metrics and Reference Methods

Three performance metrics are adopted, the first performance metric is the completion time; here, the completion time refers to the time of processing all tasks. The second performance metric is the energy consumption, which refers to the energy of processing all the tasks (for mobile terminal equipment). The third performance metric is reliability, which refers to the success rate of the task execution under the constraints of the maximum tolerance time and the residual energy available.

The proposed task scheduling strategy- HH is compared with other three strategies, which are IPSO, IACO and RR. The HH task scheduling strategy is based on the hybrid heuristic algorithm; the IPSO task scheduling strategy is based on the improved particle swarm optimization algorithm; the IACO task scheduling strategy is based on the improved ant colony optimization algorithm; the RR task scheduling strategy is based on the round-robin algorithm.

### 5.3. Evaluation Results

#### 5.3.1. Completion Time

First, the completion time is used to verify the performance of the four task scheduling strategies. Experiments were conducted 20 times, and then the average was used. The task number is set to 50, 100, 150, 200, 250 and 300 respectively. The results are shown in Figure 5.

In Figure 5, we examine how the completion time delivered by our algorithm changes with the growth of the number of tasks. With the increasing task number, the completion time of the four task’s scheduling strategies is also increasing. The completion time of HH is the smallest, the completion time of IPSO and IACO are greater than HH, and the completion time of RR is the largest. This is because HH, IPSO and IACO conduct the task scheduling by taking the minimum completion time as a goal. In addition, HH integrates the advantages of IPSO with IACO, so the performance of HH is better than that of IPSO and IACO. RR schedules task requests to different fog nodes in turn, and the completion time is not considered in the process of scheduling; therefore, the performance of RR’s completion time is the worst.

#### 5.3.2. Energy Consumption

The energy consumption simulation results of the four task scheduling strategies under different numbers of tasks are shown in Figure 6. It is clearly shown that with the increase of the number of tasks, the energy consumptions of all the four scheduling strategies present a rising trend, the reason is that with the growth of the task number, the terminal devices have to consume more energy to complete the process of sending of the task data and receiving the task results. The energy consumption of terminal device is decided by the power and the elapsed time of terminal device. The power value of the terminal device is fixed; therefore, the energy consumption is only related to the completion time, for which the more the completion time is, the more the terminal device energy consumption.

From Figure 6, it can be concluded that the energy consumption of the HH is the lowest, which is in keeping with minimum completion time of the HH. The energy consumption of the IPSO and IACO are higher than the HH. This is because the completion time of the IPSO and IACO is more than the HH. The completion time of the RR is the most, so this leads to the energy consumption of RR reaching the highest value. When the task number is less than 100, the completion time difference of the four strategies is small, so the energy consumption difference is also small. When the task number is more than 100, however, the waiting delay is increasing, where the waiting delay includes the waiting delay in the terminal equipment and the waiting delay in the fog node. Since the greater the task number is, the more the waiting delay; therefore, the task completion time will increase, and the purpose of HH is to accomplish the scheduling of different tasks to different fog nodes meanwhile make the completion time achieve a minimum, and realize the energy consumption minimum.

#### 5.3.3. Reliability

To verify the reliability of the four kinds of task scheduling strategies, the number of tasks is initially changed, the task number increases from 50 to 300, and the reliability change of the four kinds of task scheduling strategies can be observed, the simulation results are shown in Figure 7.

Then, the number of tasks is set at 200, the maximum tolerance time of the tasks is changed, increasing from 10 s to 100 s and observing the reliability of the four scheduling strategies under different maximum tolerance times. The simulation results are shown in Figure 8. Figure 7 presents the reliability change trend of four strategies under different task numbers. With the number of tasks increasing, the reliability of strategies shows an obvious downward trend. When the task number is less than 100, the change tendency of the four strategies is gentle, and the reliability is very high. This occurs because in this period of the scope of the task before the transfer, there is no waiting for the delay, and fewer tasks will not cause the computing pressure of the fog node for the fog mode; therefore, within the scope of the task number, the completion time and energy consumption are very small, which is also in in keeping with the results of the above task completion time. As seen from the diagram, the reliability of HH is the highest in the four strategies, the reason is that both completion time and energy consumption of HH are the least, then are IPSO and IACO, while the reliability of the RR is the lowest.

Figure 8 further explores the reliability performance under the proposed HH strategy and the other three scheduling strategies. With the growth of the maximum tolerance time, the reliability of the four strategies shows a rising trend, the reliability of HH always is the highest, the reliability of the IPSO and IACO is between the HH and RR, while the reliability of the RR is the lowest. In the process of task scheduling, three strategies (HH, IPSO and IACO) not only meet the requirements of real-time tasks but also to minimize the task completion time as the optimization goal. The HH concentrates the advantages of the IPSO and IACO, so the HH outperforms the IPSO and IACO.

By comprehensively considering completion time, power consumption and reliability, the comprehensive performance of the HH is the best among the four scheduling strategies, so the HH is an efficient task scheduling strategy in fog computing.

## 6. Conclusions

This paper has proposed a HH algorithm for task scheduling in smart production lines with fog computing. The smart production line system model is set up and a related mathematic model concerning task scheduling is also given. Considering the features of terminal devices, tasks and fog nodes, an objective function of task scheduling of fog computing is formulated under the constraints of delay and energy consumption. To solve the task scheduling effectively, the HH algorithm which combines the advantages of IPSO and IACO is proposed to search for the optimal solution. The HH algorithm is validated by simulation, and compared with other three algorithms, our strategy demonstrates the best performance using three performance metrics. Our future research work will extend this research on task clustering and fog node clustering for task scheduling in fog computing.

## Figures and Tables

**Figure 1 sensors-19-01023-f001:**
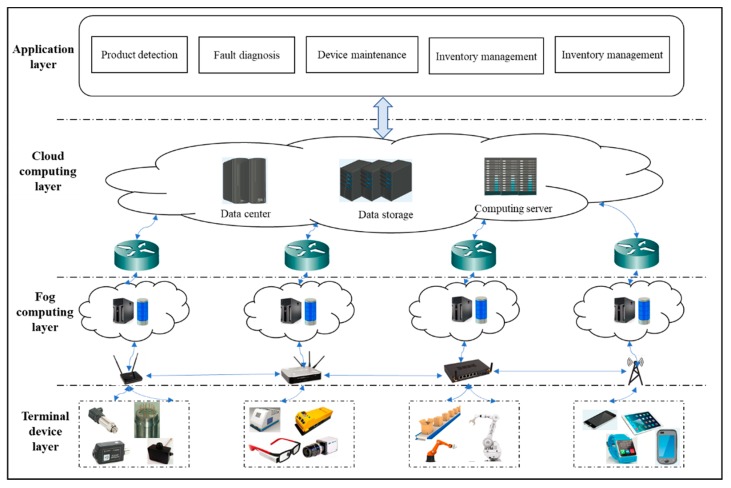
Smart production line system architecture based on fog computing.

**Figure 2 sensors-19-01023-f002:**
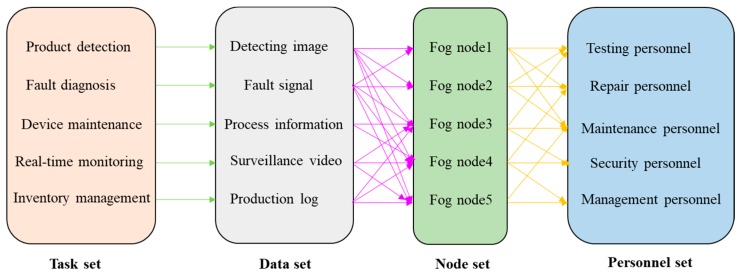
Heterogeneous task processing flow in a fog environment.

**Figure 3 sensors-19-01023-f003:**
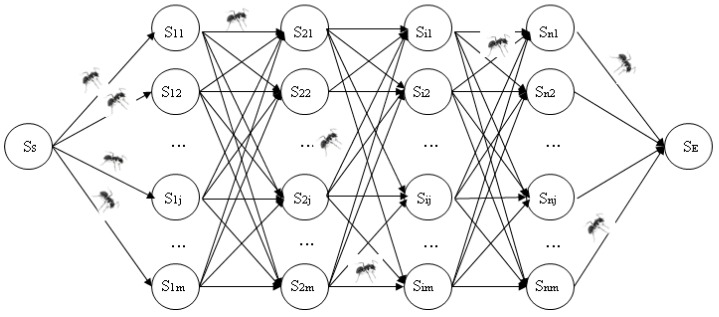
The ant colony algorithm structure towards task scheduling.

**Figure 4 sensors-19-01023-f004:**
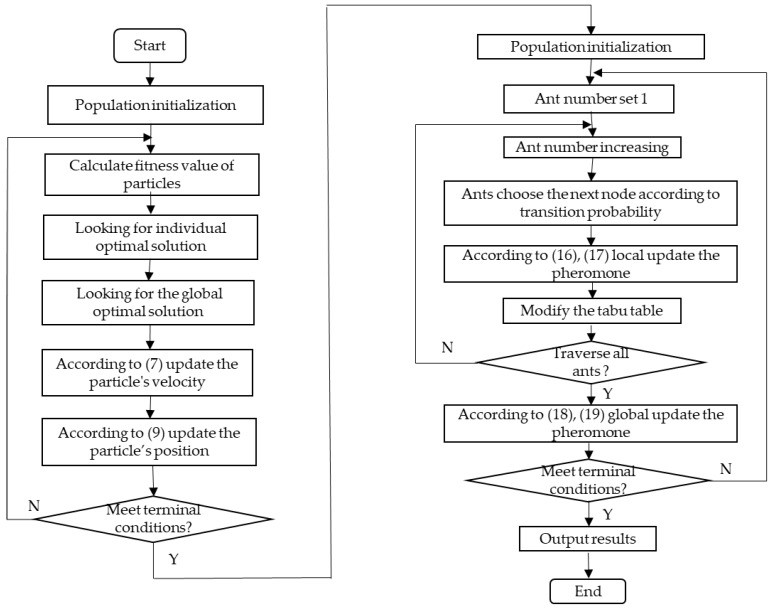
The flowchart of the hybrid heuristic algorithm.

**Figure 5 sensors-19-01023-f005:**
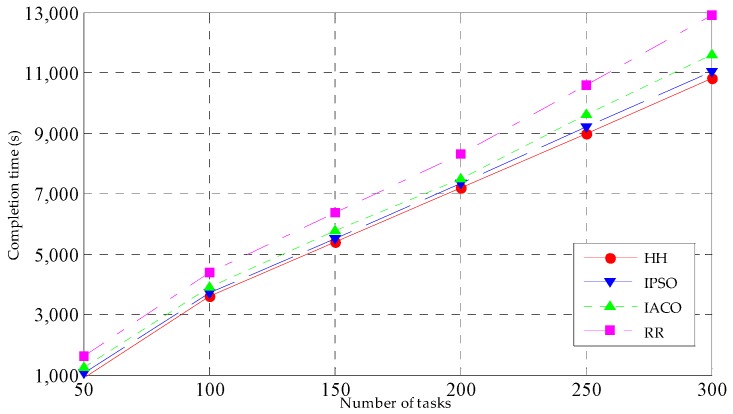
Comparison of the completion time of four strategies with different task numbers.

**Figure 6 sensors-19-01023-f006:**
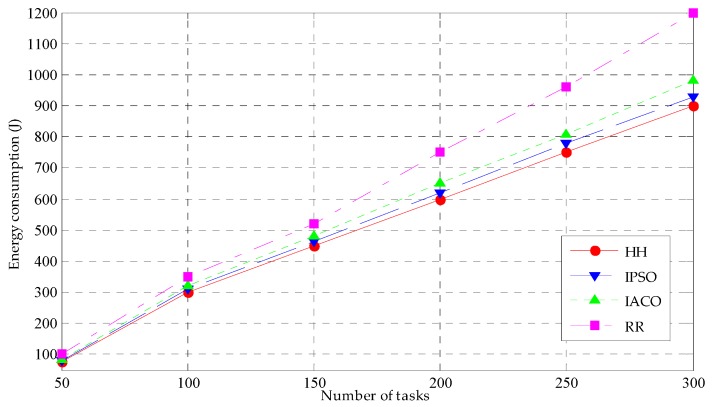
Comparison of the energy consumption of four strategies with different task numbers.

**Figure 7 sensors-19-01023-f007:**
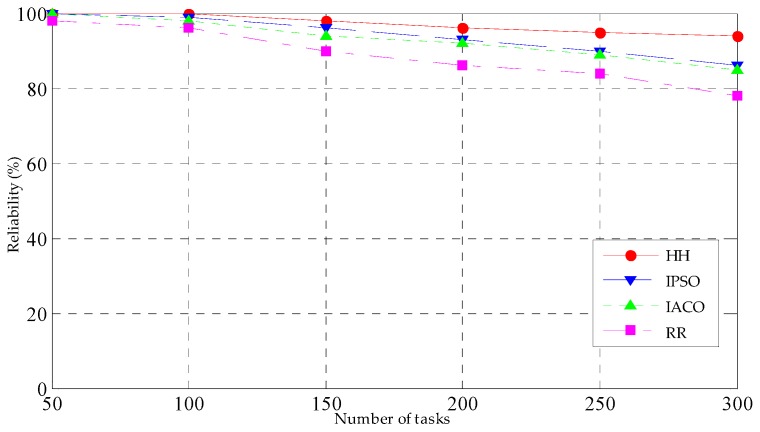
Comparison of the reliability of four strategies with different task numbers.

**Figure 8 sensors-19-01023-f008:**
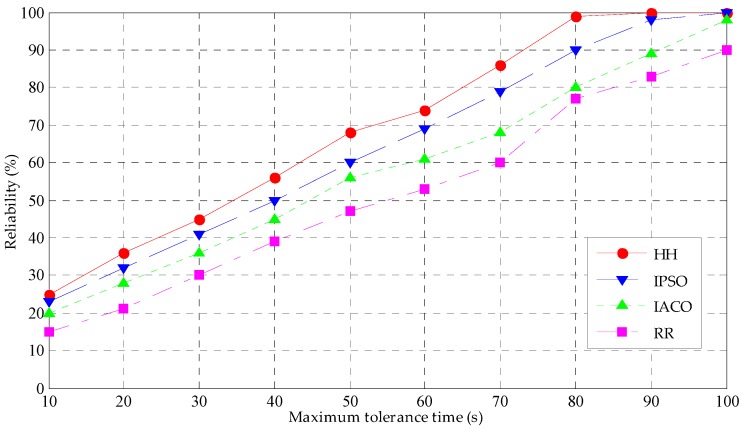
Comparison of the reliability of four strategies with different maximum tolerance times.

**Table 1 sensors-19-01023-t001:** Parameter values of the simulation.

Parameters	Valve	Description
*n*	100	number of terminal devices
*m*	10	number of fog nodes
*D_i_*	10–50 Mb	data size of *I_i_*
*C_j_*	1–2G cycles/s	computing capacity of *F_j_*
*θ_c_*	300 cycles/bit	computing intensity
*P_ir_*	0.1 W	transmission power of *g_i_*
*P_ie_*	0.05 W	idle power of *g_i_*
*B*	100 MHz	link bandwidth
*μ_T_*	0.7	real-time satisfaction weight
*μ_E_*	0.3	energy consumption satisfaction weight
*ω*	[0.4, 0.9]	inertia weight factor
*η*	0.9	constriction factor
*K_max_*	200	maximum iterations
*c*_1_, *c*_2_	2	accelerate factor
*r*_1_, *r*_2_, *r*_3_	0–1	random variable
*α*	1	pheromone weight coefficient
*β*	1	weight coefficient of heuristic information
*ρ*	0.5	pheromone volatilization coefficient

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
