# Peer review of "Task Scheduling Based on a Hybrid Heuristic Algorithm for Smart Production Line with Fog Computing"

_sensors, 2019, doi:10.3390/s19051023_

Round 1

Reviewer 1 Report

Line 42 to 48 While various references of fog computing is presented it is not clear why we are presenting the specific reference? The paper need specific reference showing what is been done in the specific area and not in Fog Computing in general.

The problem formulation in the Introduction is not clear We need to make clear what this paper is targeting and why we are using this Hybrid heuristics algorithm.

In related work we see one reference per technology so we see in 73 architecture [8] but this is not true  there are various fog architectures so maybe Survey paper reference are needed...

eg.

Naha, Ranesh Kumar, Saurabh Garg, Dimitrios Georgakopoulos, Prem Prakash Jayaraman, Longxiang Gao, Yong Xiang, and Rajiv Ranjan. "Fog Computing: survey of trends, architectures, requirements, and research directions." IEEE access 6 (2018): 47980-48009.

line 74 "and so on" is informal...

More highly respected reference are need from well known Sources e.g. IEEE Communication Magazine e.g.

Markakis, Evangelos K., Kimon Karras, Nikolaos Zotos, Anargyros Sideris, Theoharris Moysiadis, Angelo Corsaro, George Alexiou et al. "EXEGESIS: Extreme edge resource harvesting for a virtualized fog environment." IEEE Communications Magazine 55, no. 7 (2017): 173-179.

Markakis, Evangelos K., Kimon Karras, Anargyros Sideris, George Alexiou, and Evangelos Pallis. "Computing, Caching, and Communication at the Edge: The Cornerstone for Building a Versatile 5G Ecosystem." IEEE Communications Magazine 55, no. 11 (2017): 152-157.

and others.. eg.

Mutlag, Ammar Awad, Mohd Khanapi Abd Ghani, N. Arunkumar, Mazin Abed Mohamed, and Othman Mohd. "Enabling technologies for fog computing in healthcare IoT systems." Future Generation Computer Systems 90 (2019): 62-78.

Line 232 an assumption paragraph is missing.

Varios times inside the paper we see the following repeats:

in this paper, The paper, We introduce, paper.

Please be more formal in describing your work

LINE 339 Figure 3 need a clear description

LINE 287 The Heuristic algorithm miss it's math description...

The discussion of the results is missing.

The conclusion is to limited please extend and include a future work part.

Author Response

Response to Reviewer 1 Comments

Point 1: Line 42 to 48 While various references of fog computing is presented it is not clear why we are presenting the specific reference? The paper need specific reference showing what is been done in the specific area and not in Fog Computing in general.

Response 1: We thank the reviewer for the comment. Rely on its own advantages, fog computing plays a great role in the field of smart manufacturing. The performance of fault detection and state analysis of devices in production line can be improved through a wealth of computational and storage services provided by fog computing [6]. Efficient manufacture inspection system with fog computing is implemented [7]. A fog-based solution is proposed in [8] for real-time monitoring and data processing in manufacturing. By using fog computing technology an enhancing smart maintenance management solution is presented in [9]. In line 42 to 47.

Point 2: The problem formulation in the Introduction is not clear We need to make clear what this paper is targeting and why we are using this Hybrid heuristics algorithm.

Response 2: We thank the reviewer for the comment. Therefore, determining the approach of task scheduling among different fog nodes to minimize the delay and energy consumption is the purpose of this paper. As well as we know, task scheduling is an NP-Hard problem, it is also a very challenging problem. Therefore, a hybrid heuristic algorithm to solve the task scheduling problem has been a research hotspot of scholars worldwide. The hybrid heuristic algorithm which combines the advantages of a variety of heuristic algorithm, the accuracy of results and the optimization process are both improved. In line 53 to 60.

Point 3: In related work we see one reference per technology so we see in 73 architecture [8] but this is not true  there are various fog architectures so maybe Survey paper reference are needed...

eg.

Naha, Ranesh Kumar, Saurabh Garg, Dimitrios Georgakopoulos, Prem Prakash Jayaraman, Longxiang Gao, Yong Xiang, and Rajiv Ranjan. "Fog Computing: survey of trends, architectures, requirements, and research directions." IEEE access 6 (2018): 47980-48009.

Response 3: We thank the reviewer for the comment. Some references are added in line 81-83. In recent years, many scholars worldwide have conducted research on fog computing, and the main research directions are focused on the definition [13,14], architecture [15-17], application [18-20], computing offloading [21-23] and task scheduling [24-26].

Point 4: line 74 "and so on" is informal...

More highly respected reference are need from well known Sources e.g. IEEE Communication Magazine e.g.

Markakis, Evangelos K., Kimon Karras, Nikolaos Zotos, Anargyros Sideris, Theoharris Moysiadis, Angelo Corsaro, George Alexiou et al. "EXEGESIS: Extreme edge resource harvesting for a virtualized fog environment." IEEE Communications Magazine 55, no. 7 (2017): 173-179.

Markakis, Evangelos K., Kimon Karras, Anargyros Sideris, George Alexiou, and Evangelos Pallis. "Computing, Caching, and Communication at the Edge: The Cornerstone for Building a Versatile 5G Ecosystem." IEEE Communications Magazine 55, no. 11 (2017): 152-157.

and others.. eg. 

Mutlag, Ammar Awad, Mohd Khanapi Abd Ghani, N. Arunkumar, Mazin Abed Mohamed, and Othman Mohd. "Enabling technologies for fog computing in healthcare IoT systems." Future Generation Computer Systems 90 (2019): 62-78.

Response 4: We thank the reviewer for the comment. “and so on” is deleted in line 101. The references are added in line 586-597.

17.    Naha, Ranesh Kumar, Saurabh Garg, Dimitrios Georgakopoulos, Prem Prakash Jayaraman, Longxiang Gao, Yong Xiang, and Rajiv Ranjan. Fog Computing: survey of trends, architectures, requirements, and research directions. IEEE Access, 2018, 6, 47980–48009.

18.    Markakis, Evangelos K., Kimon Karras, Nikolaos Zotos, Anargyros Sideris, Theoharris Moysiadis, Angelo Corsaro, George Alexiou et al. EXEGESIS: Extreme edge resource harvesting for a virtualized fog environment. IEEE Communications Magazine 55, no. 7 (2017): 173-179.

19.    Markakis, Evangelos K., Kimon Karras, Anargyros Sideris, George Alexiou, and Evangelos Pallis. Computing, Caching, and Communication at the Edge: The Cornerstone for Building a Versatile 5G Ecosystem. IEEE Communications Magazine 55, no. 11 (2017): 152-157.

20.    Mutlag, Ammar Awad, Mohd Khanapi Abd Ghani, N. Arunkumar, Mazin Abed Mohamed, and Othman Mohd. Enabling technologies for fog computing in healthcare IoT systems. Future Generation Computer Systems 90 (2019): 62-78.

Point 5: Line 232 an assumption paragraph is missing.

Response 5: We thank the reviewer for the comment. An assumption paragraph is added in line 239-243.

To simplify the problem complexity and reduce the difficulty to solve the problem, the following assumptions are put forward: each task is independent and there is no constraint relationship among tasks; each task can only be allocated to a fog node and all the tasks are not allowed to allocate repeat. The task in the computation process does not consider the impact of the mobility of the terminal equipment. All the fog nodes are static and task in the process of execution cannot be interrupted.

Point 6: Varios times inside the paper we see the following repeats:

in this paper, The paper, We introduce, paper.

Please be more formal in describing your work

Response 6: We thank the reviewer for the comment. We are more formal in describing our work in line 78, 80, 85, 115, 262, 270, 377, 501.

Point 7: LINE 339 Figure 3 need a clear description 

Response 7: We thank the reviewer for the comment. A clear description is added in line 343-351. The total number of ants is σ, and each ant has the following features: the ants walking route is a directed acyclic graph, as shown in Figure 3, and all the ants start from the starting point for SS, then choose the next node according to the path transition probability. The path transition probability is a function of the path distance and the path pheromone amount. The legal walking route of each ant is shown in Figure 3, and to prevent the ant to have ever walked to the node, a tabu table is introduced. The tabu table of ant k is set to tabuk; after the ant k has walked to the node sij, the node sij will be added to the tabu table, and in the process where the ants are walking below it, sij will not be chosen. When the ant k walks to the end point SE, the ant has completed its travel (all task scheduling), and the ants leave pheromone on the path to walk through

Point 8: LINE 287 The Heuristic algorithm miss it's math description...

Response 8: We thank the reviewer for the comment. A math description of the heuristic algorithm is added in line 304-305. where, the objective function of task scheduling is the denominator of formula (10), when the fitness function value is greater, the solution is better.

Point 9: The discussion of the results is missing.

Response 9: We thank the reviewer for the comment. We have modified the paper title in line 435, frankly speaking, the section of evaluation results has already included the discussion of the results.

Point 10: The conclusion is to limited please extend and include a future work part.

Response 10: We thank the reviewer for the comment. The conclusion is extended and a future work is given in line 525-534.

This paper has proposed a HH algorithm on task scheduling for smart production line with fog computing. The smart production line system model is set up and related mathematic model about task scheduling are also given. Considering the features of terminal devices, tasks and fog nodes, an objective function of task scheduling of fog computing is formulated under the constraints of delay and energy consumption. To solve the task scheduling effectively, the HH algorithm which combines the advantages of IPSO and IACO is proposed to search for the optimal solution. The HH algorithm is validated by simulation, and compared with other three algorithms, our strategy demonstrates the best performance using three performance metrics. Our future research work is to extend this research on task clustering and fog node clustering for task scheduling in fog computing.

Reviewer 2 Report

Authors present new ant-colony based task scheduling for smart manufacturing based on fog computing under the constraints of delay and energy consumption. The work has a chance to be published subjected to the following comments:

In an introduction, authors should highlight the contribution in better way and stress on the heterogeneous tasks instantiated on the nodes in Fog environment, related routing models and their KPI challenges and how you deal with it which is one of appealing outcome of this solution, in relation to the existing problems and how you tackle the problem which is messy and hard to catch. 

Authors present new ant-colony based task scheduling for smart manufacturing based on fog computing under the constraints of delay and energy consumption.  The work has a chance to be published subjected to the following comments:

In an introduction, authors should highlight the contribution in better way and stress on the heterogeneous tasks instantiated on the nodes in Fog environment, related routing models and their KPI challenges and how you deal with it which is one of appealing outcome of this solution, in relation to the existing problems and how you tackle the problem which is messy and hard to catch. 

To make the conclusion section more clear, authors are highly encouraged to include the point-by-point findings of this article. The current conclusion is written very wide, and it is not easy to maintain the key findings.

I am sure that 300 nodes cannot be assumed as large scenario, so, it is expected to test your system in the much dense network. How your solution can deal with various topologies of Fog?

Figures should have a reasonably concise, precise caption to present the related figures. Besides, the structure of the algorithm needs to be detailed.

How your local bio-inspired solution can be applied to such a system? Why ANT? Why you do not the select global solution?

How is the complexity of heuristics against the optimal case in large scale nodes with various users and demands? Can you prove it?   

The results should be supported with related explanations. Also, results figures are unclear and not easy to understand.

I found background is limited, and the related in-scope works may be added  like ”Fog of everything: Energy-efficient networked computing architectures, research challenges, and a case study”

To make the conclusion section more clear, authors are highly encouraged to include the point-by-point findings of this article. The current conclusion is written very wide, and it is not easy to maintain the key findings.

Author Response

Response to Reviewer 2 Comments

Point 1: In an introduction, authors should highlight the contribution in better way and stress on the heterogeneous tasks instantiated on the nodes in Fog environment, related routing models and their KPI challenges and how you deal with it which is one of appealing outcome of this solution, in relation to the existing problems and how you tackle the problem which is messy and hard to catch. 

Response 1: We have highlight the contribution in line 61. The heterogeneous tasks are instantiated in figure 2. The related system models and mathematics models is given in line 155 and 211. The KPI challenges are delay and energy consumption, by executing task scheduling among fog nodes to reduce delay and energy consumption, a heuristic algorithm is used to solve the task scheduling objective function.

Point 2: To make the conclusion section more clear, authors are highly encouraged to include the point-by-point findings of this article. The current conclusion is written very wide, and it is not easy to maintain the key findings.

Response 2: The conclusion is modified in line 525-534. This paper has proposed a HH algorithm on task scheduling for smart production line with fog computing. The smart production line system model is set up and related mathematic model about task scheduling are also given. Considering the features of terminal devices, tasks and fog nodes, an objective function of task scheduling of fog computing is formulated under the constraints of delay and energy consumption. To solve the task scheduling effectively, the HH algorithm which combines the advantages of IPSO and IACO is proposed to search for the optimal solution. The HH algorithm is validated by simulation, and compared with other three algorithms, our strategy demonstrates the best performance using three performance metrics. Our future research work is to extend this research on task clustering and fog node clustering for task scheduling in fog computing.

Point 3: I am sure that 300 nodes cannot be assumed as large scenario, so, it is expected to test your system in the much dense network. How your solution can deal with various topologies of Fog?

Response 3: We are very agree with your view that 300 nodes cannot be assumed as large scenario, so, in this paper we take a smart production line as an application scenario, 300 nodes are enough for the smart production line, and software defined network (SDN) technology is applied to deal with various topologies of Fog.

Point 4: Figures should have a reasonably concise, precise caption to present the related figures. Besides, the structure of the algorithm needs to be detailed.

Response 4: The precise captions are presented in line 164, 191, 353, 412, 463, 483, 502, 514. The structure of the algorithm is detailed.

Figure 1. Smart production line system architecture based on fog computing.

Figure 2. Heterogeneous tasks processing flow in fog environment

Figure 3. The ant colony algorithm structure towards task scheduling

Figure 4. The flowchart of the hybrid heuristic algorithm

Figure 5. Comparison of the completion time of four strategies with different task numbers

Figure 6. Comparison of the energy consumption of four strategies with different task  numbers

Figure 7. Comparison of the reliability of four strategies with different task numbers

Figure 8. Comparison of the reliability of four strategies with different maximum tolerance times

Point 5: How your local bio-inspired solution can be applied to such a system? Why ANT? Why you do not the select global solution?

Response 5: The optimization result of ant colony algorithm has higher precision, in this paper, the ant colony algorithm is improved, the ants not only after each loop to a global update of path, but the path construction as well as local update.

Point 6: How is the complexity of heuristics against the optimal case in large scale nodes with various users and demands? Can you prove it?

Response 6: The time complexity of heuristics is O(n^4), time = n*(n-1)*m*T/2, where n is the number of tasks, m is the number of ants or particles, T is the number of iterations, usually m=n*2/3, T=k*n. so, time=n*(n-1)*n*n*k/3, when n->∞, time ≈n^4.

 The space complexity of heuristics is O(n^2), space = 3*n*n+n*n*2/3, when n->∞, space ≈n^2.

Point 7: The results should be supported with related explanations. Also, results figures are unclear and not easy to understand. 

Response 7: Some related explanations of the results are added in section of evaluation results.

Point 8: I found background is limited, and the related in-scope works may be added  like ”Fog of everything: Energy-efficient networked computing architectures, research challenges, and a case study”

Response 8: The references are added in line 577-578 and line 586-597.

14.    Baccarelli, E.; Naranjo, P G V.; Scarpiniti, M.; Abawajy, J H. Fog of everything: Energy-efficient networked computing architectures, research challenges, and a case study. IEEE Access, 2017, 5, 9882–9910.

17.    Naha, Ranesh Kumar, Saurabh Garg, Dimitrios Georgakopoulos, Prem Prakash Jayaraman, Longxiang Gao, Yong Xiang, and Rajiv Ranjan. Fog Computing: survey of trends, architectures, requirements, and research directions. IEEE Access, 2018, 6, 47980–48009.

18.    Markakis, Evangelos K., Kimon Karras, Nikolaos Zotos, Anargyros Sideris, Theoharris Moysiadis, Angelo Corsaro, George Alexiou et al. EXEGESIS: Extreme edge resource harvesting for a virtualized fog environment. IEEE Communications Magazine 55, no. 7 (2017): 173-179.

19.    Markakis, Evangelos K., Kimon Karras, Anargyros Sideris, George Alexiou, and Evangelos Pallis. Computing, Caching, and Communication at the Edge: The Cornerstone for Building a Versatile 5G Ecosystem. IEEE Communications Magazine 55, no. 11 (2017): 152-157.

20.    Mutlag, Ammar Awad, Mohd Khanapi Abd Ghani, N. Arunkumar, Mazin Abed Mohamed, and Othman Mohd. Enabling technologies for fog computing in healthcare IoT systems. Future Generation Computer Systems 90 (2019): 62-78.

Round 2

Reviewer 2 Report

The updated version notably addressed my comments. I feel the work is fine for the journal

This manuscript is a resubmission of an earlier submission. The following is a list of the peer review reports and author responses from that submission.

Round 1

Reviewer 1 Report

This paper considers a static scheduling of tasks to fog nodes. Many unrealistic assumptions that simplify the system, execution and the tasks models are made. For example, a very simple task is used. The execution time on all the fog nodes are known a priori. Also, once a task is allocated to a node, it will execute on that node. A node can only execute one task at a time. The task type used in the paper is not explained. The symbols used are not defined and quite confusing the way it is used. There also duplicate use of symbol ‘g’ for two different entities.  Also the values of energy for the nodes are arbitrarily set, which an indicative that this paper needs some more time to mature to a level that can be submitted for a publication. The simulation does not consider the time taken by PSO+ ACO. The major failure of the work is that the authors indicate that the environment is for processing real-time tasks. However, what is described and the scheduling algorithm discussed are not real time but a static scheduling where everything including execution of the tasks on all fog nodes are known. In fact, the authors simply describe the standard scheduling algorithm for conventional distributed systems (e.g., Grid, Cluster, Cloud) to the fog environment (fog nodes).

The authors indicate that “strong fog nodes have a lot of computing resources, they can help the weak fog nodes to complete the task processing.” However, tasks are not allowed to move once assigned and this claim is wrong.

Half of the paper is on explanation of some concepts as definition 1 to 7. Most of these definitions can be presented compactly in fewer paragraphs. A wide variety of terminologies are used to mean “task completion time” rather wrongly. What does “time cost” and “energy consumption cost” mean? How is “Task completion time” differ from “time cost”? You did not define the “Task scheduling reliability” concept. It is my understanding that the terminal equipment does not do the processing of the task. The question is why you consider the energy consumption of the terminal equipment?

The language is a bit difficult to understand. Example of spelling “fog computing mode” which should be “fog computing node.” Example of grammatically incorrect sentences is the “Statistics the number of failed tasks denote …..” and “it is difficult to through the polynomial method to find the optimal solution.” New problem appearing, how to carry on the task scheduling among the different fog nodes, and makes the time cost and energy consumption cost minimum.” Example of incomprehensible sentences is the task in the process does not consider the impact of the mobility of the terminal equipment”. I could not find information on “task in the process”. I also could not understand what it means the task does not consider the impact of mobility. There are many such confusing statements throughout the paper. The authors do not define what Terminal equipment is.